# Iridoids and Amino Acid Derivatives from the Paraguayan Crude Drug *Adenocalymma marginatum* (ysypó hû)

**DOI:** 10.3390/molecules25010180

**Published:** 2020-01-01

**Authors:** Guillermo Schmeda-Hirschmann, Alberto Burgos-Edwards, Felipe Jiménez-Aspee, Daniel Mieres-Castro, Cristina Theoduloz, Lisa Pormetter, Ramon Fogel, Claudia Céspedes, Nelida Soria, Sintya Valdez

**Affiliations:** 1Laboratorio de Química de Productos Naturales, Instituto de Química de Recursos Naturales, Universidad de Talca, Campus Lircay, Talca 3460000, Chile; aburgos@utalca.cl (A.B.-E.); dmieres@utalca.cl (D.M.-C.); l.pormetter@tu-braunschweig.de (L.P.); 2Departamento de Ciencias Básicas Biomédicas, Facultad de Ciencias de la Salud, Universidad de Talca, Campus Lircay, Talca 3460000, Chile; fjimenez@utalca.cl; 3Laboratorio de Cultivo Celular, Facultad de Ciencias de la Salud, Universidad de Talca, Campus Lircay, Talca 3460000, Chile; ctheodul@utalca.cl; 4Centro de Estudios Rurales Interdisciplinarios, Oliva 1019, Edificio Lider V, Piso 17, oficina 172, Asunción 01421, Paraguay; ceripy@gmail.com (R.F.); ccespedes760@gmail.com (C.C.); nsoria2000@yahoo.com (N.S.); sintyavaldez@gmail.com (S.V.)

**Keywords:** *Adenocalymma marginatum*, Bignoniaceae, counter-current chromatography, iridoids, amino acid derivatives, Guarani medicine, Paraguayan crude drug

## Abstract

The crude drug ysypó hû (*Adenocalymma marginatum* DC., Bignoniaceae) is used traditionally by the Guarani of Eastern Paraguayan as a male sexual enhancer. The aim of the present study was to identify the main constituents of the crude drug and to evaluate the in vitro inhibitory activity towards the enzyme phosphodiesterase-5 (PDE-5). The main compounds were isolated by counter-current chromatography (CCC). The metabolites were identified by spectroscopic and spectrometric means. The chemical profiling of the extracts was assessed by high-performance liquid chromatography coupled to mass spectrometry (HPLC-MS/MS). The crude extract and main isolated compounds were tested for their PDE-5 inhibitory activity using commercial kits. The iridoid theviridoside and 4-hydroxy-1-methylproline were isolated as the main constituent of the crude drug. Four chlortheviridoside hexoside derivatives were detected for the first time as natural products. Chemical profiling by HPLC-MS/MS led to the tentative identification of nine iridoids, six phenolics, and five amino acids. The crude extracts and main compounds were inactive towards PDE-5 at concentrations up to 500 µg/mL. Iridoids and amino acid derivatives were the main compounds occurring in the Paraguayan crude drug. The potential of ysypó hû as a male sexual enhancer cannot be discarded, since other mechanisms may be involved.

## 1. Introduction

The Guarani Amerindians inhabited large parts of eastern South America before the European conquest. The group included the Tupi and Guarani people, living from the costs of the Atlantic Ocean to the subtropical forests of Paraguay and eastern Argentina. The Tupi–Guarani Amerindians are closely related, sharing a common language and playing a relevant role in the South American traditional medicine and culture [1]. Paraguayan traditional medicine has strong roots in Guarani culture and has provided the world with relevant food and medicinal plants, including the tea mate (*Ilex paraguariensis*) [2], the sweet herb stevia (“kaa hee”, *Stevia rebaudiana*) [3], and hundreds of medicinal plants widely used in South America. 

The use of medicinal plants by the Guarani people was described by Pedro de Montenegro (1770), during the Spanish Jesuit Missions in Paraguay [4]. Plant and animal drugs were selected based on empirical knowledge and the self-experience of the healers. The use of sexual enhancers by the Guarani people is not common. However, fertility control was widespread, with plants used as contraceptive and as abortifacients [5]. The Bignoniaceae *Adenocalymma marginatum* (Cham.) DC. (Figure 1) is commonly known as “ysypó hû” (black vine) by the Pai Tavytera and Mbyá Amerindians, who live in northeastern Paraguay. The plant has the synonyms *Adenocalymma marginatum* var. marginatum, *A. marginatum* var. polystachyum DC, and the basionym *Bignonia marginata* Cham. [6]. The plant, fresh or in decoction, is recommended to improve fecundity and as a male sexual enhancer [7,8]. The crude drug includes the aerial parts of the plant (leaves and stem) but consists mainly of stems. The root was included in this study for comparison purposes. For the preparation of the drug, the stem and leaves are powdered, and a handful is placed in 1 L water and boiled for 3–5 min. After cooling, the decoction is taken orally (about 200 mL) two times a day, as long as necessary. The preparation is also included in mate tea, where mate is the carrier of the medicinal plants [2]. It is also used to relieve rheumatism and as anti-diarrheic [7,8].

The use of plants and animal parts as sexual enhancers is a known practice worldwide [9]. Since ancient times, several natural products have been used as sexual enhancers for the treatment of erectile dysfunction, including yohimbine from *Pausinystalia yohimbe* (Rubiaceae), forskolin from the Indian herb *Plectranthus barbatus* (Lamiaceae), or the “Peruvian Viagra” *Lepidium meyenii* (Brassicaceae), among many others [10]. They are popularly considered safe and “natural”, and no strict regulation is applied to control their sale. Unfortunately, a study published in 2015 showed that 61% of a total of 150 supplements were altered with synthetic phosphodiesterase-5 (PDE-5) inhibitors, including the commercial drug sildenalfil, or structurally modified analogs [11]. We recently reported the chemical composition and potential male sexual enhancer activity of *Campsiandra angustifolia*, *Minquartia guianensis*, *Swartzia polyphylla*, and *Tynanthus panurensis*, as well as from two traditional preparations of the Peruvian Amazon. In our experiment, *C. angustifolia* was found to be a potent inhibitor of PDE-5 in vitro [12]. 

The occurrence of the iridoids ipolamiide, strictoloside, and theviridoside in leaves of *A. marginatum* from Rio Grande do Sul (southern Brazil) has been reported [13]. However, no information is available on the chemical composition of the Paraguayan crude drug, nor on the male sexual enhancement capacity of *A. marginatum*. Hence, the aims of the present study were to isolate and characterize the main secondary metabolites from the Paraguayan crude drug ysypó hû. Additionally, the potential male sexual enhancer capacity was evaluated by the in vitro inhibitory capacity of this traditional preparation towards the enzyme phosphodiesterase-5.

## 2. Results and Discussion

### 2.1. CCC Isolation of the Main Compounds from Ysypó Hû

The TLC analysis of the crude drug extracts (Table 1 and Figure 2) showed that the main compounds were UV inactive and were visualized only after spraying with *p*-anisaldehyde/sulfuric acid and heating. The semi-purified fractions 8–15 from the Sephadex of the stem extract were worked up to obtain the main compounds using counter-current chromatography (CCC). The solvent system selected for the CCC was made of TBME/BuOH/ACN/H_2_O 1:3:1:5 (*v*/*v*/*v*/*v*), supplemented with 0.1% trifluoroacetic acid. As the main compounds from the crude drug were UV inactive, partition ratios (*K*_D_) values were calculated by visualizing the compounds after spraying with *p*-anisaldehyde–sulfuric acid and scanning the TLC plates for densitometric analysis. The *K*_D_ values of the different solvent systems and a figure of the original TLC used are shown in Table 1. The theoretical *K*_D_ value for the main spot in the selected solvent system was 0.21. The system was selected due to its good stability, short settling time in the presence of the sample, and the low *K*_D_ value, which allowed the separation of the compound with a low consumption of solvents. After setting the CCC in the selected conditions, 70.3% retention of the stationary phase was observed. The retention volume (V_R_) was calculated following the equation of Ito [14], which gave a V_R_ of 222 mL for the main peak. After the sample was injected, the detection of the main compound in TLC analysis began at Tube 62 (Figure 2). Considering that the pump was set at a flow rate of 4 mL/min, and the recollection was 1 min/tube, the experimental *K*_D_ value of the main compound was 0.28, in agreement with the theoretical calculation. Fractions from the CCC separation were pooled according to similarities of the TLC profiles. 

The main compound found in fractions 62–71 (103 mg) of the CCC was a colorless solid, soluble in methanol and insoluble in chloroform. The ^1^H-NMR spectrum of the compound showed two singlets at δ 7.52 and δ 5.72 ppm, suggesting a conjugated double bond and a trisubstituted double bond. In addition, a doublet at δ 5.55 and a second doublet at δ 4.65 agreed with a hemiacetal and an anomeric H, respectively. Two geminal coupling doublets at δ 4.25 and δ 4.14 ppm indicated a primary alcohol, and the singlet at δ 3.75 agreed with a methyl ester, supporting an iridoid skeleton.

The ^13^C-NMR spectrum showed 17 C signals, including 2 trisubstituted double bonds, a carboxylic acid methyl ester at δ 166.96 and 50.41 ppm, and 6 C signals assignable to a hexose. The structure was established following the heteronuclear multiple bond correlation (HMBC) experiments. The mass spectrum of the sodium adduct (calculated 427.1216, found 427.1196) indicated a molecular formula of C_17_H_24_O_11_, in full agreement with the iridoid glucoside theviridoside [15,16]. The acetylated fraction pool 8–15 afforded the disaccharide sucrose (as octaacetate) and theviridoside pentaacetate after purification by column chromatography (Figure 3).

Theviridoside. Molecular formula: calculated for C_17_H_24_O_11_: 404.1319; found: 404.1325; theviridoside pentaacetate: 614.1861 (calculated for C_27_H_34_O_16_: 614.1847). The NMR data of theviridoside and its pentaacetate are summarized in Table 2.

The fractions 58–60 (50 mg) from the CCC separation contained a compound showing in the FT-IR spectrum absorption bands of a carboxylic acid function at 1682 cm^−1^ and hydroxyl at 3369 cm^−1^. The ^1^H-NMR spectrum presented a methyl singlet at δ 3.08, compatible with a *N*-methyl group, a dd at δ 4.36 (1H), a m at δ 4.55 (1H), and two pairs of geminal coupling H forming a CH-CH_2_-CH-CH_2_ sequence. The ^13^C-NMR spectrum indicated six C, one of them for the carboxylic acid at δ 170.38, a *N*-methyl group at δ 42.79, two triplets for the CH_2_ groups, and two doublets, assigned by correlation spectra as the -CHOH and α-amino acid CH from a small heterocyclic ring. The compound was identified as 4-hydroxy-1-methyl proline (compound I), in agreement with the molecular formula C_6_H_11_NO_3_ calculated from the positive ion mass spectrum. The structure of the compound is shown in Figure 3 and the NMR data are summarized in Table 3. The compound has been previously reported as a stress response metabolite from *Tamarix* species [17] and from the toxic plant *Ipomoea carnea* [18]. Fractions 65–70 (24 mg) afforded a mixture of monosaccharides and the amino acids isoleucine and proline.

4-hydroxy-1-methyl proline. Molecular formula: C_6_H_11_NO_3_. Calculated for C_6_H_11_NO_3_: 145.0739, found (QTOF, positive ion mode): 146.0584. 146.0584 (70), 100.0665 (100), 82.0639 (32) C_5_H_9_NO: 99.0684; C_5_H_7_N: 81.0578 FT-IR (KBr, film): 3369, 3328, 1682, 1627, 1401, 1346, 1196.

### 2.2. Chemical Profiling of the Crude Drug

Little is known of the chemistry of *Adenocalymma* species. C-4 carboxyl iridoids were identified in the leaves of a Brazilian sample of *A. marginatum* [13]. In the Paraguayan crude drug, CCC isolation and HPLC-MS analyses allowed the identification of iridoids, phenolics, and amino acids, most of which were reported for the Brazilian sample. The HPLC-MS chromatogram is presented in Figure 4. The identification of the constituents and the occurrence in stem and root (plant parts used in Guarani medicine) is shown in Table 4.

#### 2.2.1. Iridoid Glycosides

The main compound of the Paraguayan crude drug was isolated and fully identified as theviridoside (**8**) (Table 2, Figure 3). Theviridoside presents a [M − H]^−^ ion at *m*/*z* 403 amu (atomic mass units), and shows a characteristic fragment at *m*/*z* 241 due to the neutral loss of hexose. This fragmentation pathway has been described previously [19]. Theviridoside was previously reported in the roots of the Apocynaceae *Thevetia peruviana* [15,16]. Related compounds have been previously identified in the Rubiaceae *Aitchisonia rosea* [20] and *Genipa americana* [21]. Minor constituents were identified or tentatively identified by examination of the fragmentation patterns in MS^n^, as well as by comparison with literature [22]. Compounds **2**, **5**, and **10** showed the iridoid base peak at *m*/*z* 241, differing in the identity and placement of the substituents. Compounds **2** and **5** showed a common base peak ion at *m*/*z* 727 amu, with the consecutive neutral loss of three hexoses being tentatively identified as theviridoside dihexosides. Compound **10** showed a base peak ion at *m*/*z* 565, with a neutral loss of 324 amu, leading to the fragment at *m*/*z* 241 amu. The compound was identified as theviridoside hexoside. The compounds **3**, **4**, **6**, and **9** showed the loss of 36 amu, leading to the base ion at *m*/*z* 727 amu for compound **3** and *m*/*z* 565 amu for compounds **4**, **6** and **9**, respectively. The tandem mass spectrometry (MS^2^) fragmentation showed the same pattern as theviridoside dihexoside or hexoside, suggesting derivatives of theviridoside. The [M − H]^−^ ions of compounds **3**, **4**, **6**, and **9** showed two peaks separated by two atomic mass units in a 3:1 ratio, in agreement with the isotopic peaks of ^35^Cl (75.8%) and ^37^Cl (24.2%), supporting the presence of a chlorine atom in the compounds.

Compounds **4**, **6**, and **9** were tentatively assigned as chlortheviridoside hexoside derivatives, while compound **3** was assigned as a chlortheviridoside dihexoside derivative. The different Rts suggested either differences in the identity of the hexoses and/or placement of the hexose in the iridoid core. The placement of the chlorine atom in the compounds remains to be established. In iridoids, it is common to find the chlorine atom placed at C-7 [23]. Examples include mentzefoliol and glucosylmentzefoliol from *Mentzelia cordifolia* (Loasaceae), stegioside I from the Lamiaceae *Physostegia virginiana* ssp. *virginiana* (Lamiaceae), and 6-*O*-*p*-hydroxybenzoylaystasioside and glutinoside derivatives from Catalpa fructus (Bignoniaceae), as well as in Asystasiosides from *Premna subscandens* (Verbenaceae) [23]. To the best of our knowledge, compounds **3**, **4**, **6**, and **9** have not been reported previously as natural products.

Compound **7** presented a molecular ion at *m*/*z* 659 amu, and the neutral loss of 256 amu and a hexose (162 amu), leading to the MS2 ions at *m*/*z* 402 and 241 amu, in agreement with theviridoside. The first neutral loss exceeded by 14 amu the molecular weight of theviridoside aglycon (242 amu). Thus, compound **7** was assigned as a methyl theviridoside aglycon–theviridoside dimer. Related compounds have been described from *Saprosma scortchenii* (Rubiaceae) [24,25].

Related compounds, including iridoid glycoside polymers, have been described from *Eucommia ulmoides* and showed antioxidant activity in vitro [26]. Dimeric iridoid glycosides linked to a sugar and/or phenyl moiety have been described for *Globularia trichosantha* [27]. Iridoids have been reported from Bignoniaceae species including *Campsis chinensis* [28] and *Pithechoctenium crucigerum* [29].

#### 2.2.2. Phenylpropanoid Glycosides

Compound **1** presented a molecular ion at [M − H]^−^ 341 amu, with the neutral loss of an hexose (−162 amu), leading to a MS2 ion at *m*/*z* 179 amu, in agreement with caffeic acid. The compound was tentatively identified as caffeoyl hexoside. Compound **11** presented a [M − H]^−^ ion at *m*/*z* 625, fragments to a base peak at *m*/*z* 461 amu, and a secondary ion at *m*/*z* 315 amu. The compound is related to acteoside/isoacteoside, differing in the dihydrocaffeoyl unit linked to the glucose core. Compound **12** presented a [M − H]^−^ ion at *m*/*z* 769 amu and the neutral loss of 162 amu, leading to the base peak at *m*/*z* 607. The further loss of a rhamnose and the ion at *m*/*z* 461 is in agreement with angoroside B or its isomer. Angoroside B has been reported as a constituent of *Scrophularia scopolii*, *Populus trichocarpa*, and *P. deltoides* with antibacterial activity [30]. Compounds **13** and **14** showed the neutral loss of a caffeoyl (−162 amu) and a rhamnose (−146 amu), leading to the fragment ions at *m*/*z* 461 and 315 amu, in agreement with acteoside (verbascoside) isomers. According to References [31,32], acteoside elutes first, followed by isoacteoside. Both compounds differ in the relative placement of the caffeoyl moiety in the glucose core, being at 4-O for acteoside and 6-O for isoacteoside. The caffeoyl–phenethyl glycosides acteoside and isoacteosides have been described as antineoplastic agents [33]. Compound **15** presented a [M − H]^−^ ion at *m*/*z* 515 and fragment ions at *m*/*z* 353 and 173, in agreement with dicaffeoylquinic acid [34]. The exact placement of the caffeic acid moieties in the compound remains to be established.

#### 2.2.3. Amino Acids and Derivatives 

HPLC-MS/MSn analysis of the extract in the positive ion mode allowed the identification of nitrogen-containing compounds. The occurrence of compound **I** was confirmed by the MS analysis, in full agreement with the literature [17,18,35]. The compound has been previously reported in *Sideroxylon obtusifolium* (Sapoteaceae), a medicinal species from Northeast Brazil with anti-inflammatory [36] and wound healing properties [37]. The authors reported that the compound at doses ranging from 25–100 mg/kg decreased licking time and paw edema in the formalin test, as well as a complete reversion of the increased number of polymorphonuclear cells in the inflammation point. Additionally, the compound decreased the expression of iNOS, TNF-α, COX-2, and NF-κβ [36]. Compounds **II** and **III** presented [M + H]^+^ ions at *m*/*z* 99.6 and 81.5 amu, respectively. Compound **II** showed the neutral loss of water (−18 amu), leading to the base peak at *m*/*z* 81.5 amu, supporting its close relationship with compound **III**. They were assigned as 1-methyl-2,3-dihydro-3-hydroxypyrrol and 1-methylpyrrol, respectively. The compounds IV presented a molecular ion at [M + H]^+^
*m*/*z* 115.7 amu. The neutral loss of 46 amu (H_2_O + CO) led to the MS2 ion at *m*/*z* 70 amu. The fragmentation pattern was in agreement with the cyclic amino acid proline [38]. Compound **V** presented a molecular ion at [M + H]^+^
*m*/*z* 132.1 amu. The neutral loss of 46 amu (H_2_O + CO) led to the MS2 ion at *m*/*z* 86.1 amu. The consecutive loss of −17 amu (NH_3_) was in full agreement with the branched amino acid isoleucine [38]. The HPLC traces are depicted in Figure 4.

The chemistry of the Paraguayan *A. marginatum* differs from that of *Adenocalymma* species previously investigated by the absence of sulfur-containing compounds and the occurrence of iridoids, phenylpropanoid glycosides, and amino acid derivatives in ysypó hû. Bioactivity studies on *Adenocalymma* have included a screening of the antimicrobial effect of climbers from the Bignoniaceae family [39] and the influence of aqueous and ethanol extracts of *Pseudocalymma alliaceum* on hematological parameters in rats [40].

### 2.3. Effect of the Extracts and Main Constituents on Phosphodiesterase-5 (PDE-5) Activity

The Paraguayan crude drug ysypó hû is traditionally used as a male sexual enhancer by the Guarani people. The root of the Bignoniaceae *Anemopaegma arvense* (Vell.) Stellfeld & J.F. Souza (catuaba) is traditionally used as an aphrodisiac in central Brazil [41]. PDE-5 is the predominant cGMP-metabolizing enzyme that is expressed in the cavernosal tissue and in penile arteries, and it is the pharmacological target of sildenafil (Viagra^®^), vardenafil (Levitra^®^), and tadalafil (Cialis^®^). These three drugs are clinically used for the treatment of erectile dysfunction because they are potent and selective PDE-5 inhibitors, which results in increased levels of cGMP in the smooth muscle cells of the penile corpus cavernosum, a decrease in the intracellular calcium, and desensitization of proteins to the effects of calcium, producing smooth muscle relaxation and penile erection [42]. The ysypó hû extracts and main compounds were assessed as potential PDE-5 inhibitors. Under our experimental conditions, neither the crude root, stem extracts, nor the main compounds presented inhibitory activity at concentrations up to 500 µg/mL. Under the same experimental conditions, sildenafil citrate showed an IC_50_ of 21.07 ± 0.61 µg/mL. Despite the lack of inhibition of PDE-5 by ysypó hû extracts and the main constituents, other mechanisms for sexual enhancement cannot be ruled out. Several natural products have demonstrated effectiveness against erectile dysfunction [43]. The indole alkaloid yohimbine from *Corynanthe yohimbe* acts as an antagonist on α2-adrenergic receptors, increasing sympathetic outflow to the periphery, increasing vasomotor activity, and creating an aphrodisiac sensation [44]. The iridoid glycosides morroniside and loganin, from the traditional chinese medicine Xiao Ke Zheng, protected against diabetes-mellitus-induced testicular damage in rats by increasing sex hormone levels (GnRH, LH, and FSH) and testicular/body weight ratio, as well as decreasing oxidative stress and cellular apoptosis in the testis [45]. The aphrodisiac properties of the Andean root of *Lepidium meyenii* (maca) have been widely described in literature [46,47]. In humans, the extract significantly improved sexual desire compared to placebo, independently of serum testosterone levels [48]. Despite all the evidence, the mechanism of action of *L. meyenii* is not clear so far. Thus, despite the lack of inhibition presented by *A. marginatum* towards PDE-5, further investigations are needed in order to clarify its potential as a male sexual enhancer.

## 3. Materials and Methods

### 3.1. Plant Material

The stems and roots of *Adenocalymma marginatum* DC. were collected on December of 2015 at Mondaymi, Distrito de San Joaquin, Departamento de Caaguazú, Paraguay (25°9’21,39”S, 56°4’44,16”W). A voucher herbarium specimen (Céspedes 1628) was deposited at the Herbario FCQ, Facultad de Ciencias Químicas, Universidad Nacional de Asunción, Asunción, Paraguay. The air dried plant material was powdered and extracted three times with EtOH:H_2_O (70:30, *v*/*v*). The combined extracts were evaporated under reduced pressure and lyophilized to yield the crude extract. The *w*/*w* extraction yields were 18% and 23% for the stem and roots, respectively. The EtOH:H_2_O (70:30) extract was preferred over the infusion for the higher extraction yields. 

### 3.2. Isolation of the Main Compounds

The lyophilized extracts of stems and roots were worked with separately in order to obtain fractions and to identify the constituents of the crude drug. To remove fats and chlorophylls, the root and stem extract were partitioned against CHCl_3_. The root extract (53.9 g) yielded a CHCl_3_-soluble (1.5% *w*/*w* yield) and a CHCl_3_-insoluble fraction (98.5% *w*/*w*). The stem extract (95.56 g) yielded a CHCl_3_-soluble (0.95% *w*/*w*) and a CHCl_3_-insoluble (99.05% *w*/*w*) fraction. The root and stem CHCl_3_-insoluble extracts, containing the polar constituents, were compared by TLC (silica gel) using different solvent mixtures as mobile phase. Both plant parts presented similar profiles (data not shown).

A total 5.5 g of the polar, CHCl_3_-insoluble stem extract was loaded into a Sephadex LH-20 column (column length: 105 cm; internal diameter: 5.2 cm; Sephadex load: 45 cm, Sigma-Aldrich, St. Louis, MO, USA) previously equilibrated with MeOH. The procedure was carried out two times in order to separate a total of 11 g of the stem extract. In both cases, the void volume was 200 mL. Thirty fractions of 60 mL each were collected. Fractions were pooled according to the TLC analysis (Silica gel; EtOAc: formic acid (98–100%): H_2_O:acetone 10:2:3:1) as follows: Fractions 1–6 did not contain compounds of interest and were discarded; Fraction 7 (225 mg) contained a main compound with Rf 0.08; Fractions 8–15 (9.16 g) showed in TLC two compounds with Rf. 0.08 and 0.27; Fractions 16–18 (320 mg) contained a main compound with Rf 0.27. Both compounds were revealed as blue spots after spraying the TLC plate with anisaldehyde and heating. Fractions 19–30 did not contain compounds of interest and were discarded. Fractions 8–15 were pooled and further purified by counter-current chromatography, as described in the following paragraphs. A total 1 g of the pooled Fractions 8–15 was acetylated (pyridine/acetic anhydride, 36 h, room temperature) and the major compound was purified by column chromatography (68 cm length, ID: 3.2 cm) on silica gel (88 g), using a PE:EtOAc 1:1 v/v mixture as eluent (isocratic). Some 170 fractions of 10 mL each were collected and combined in different pools according to the chromatographic behavior on TLC (silica gel, PE:EtOAc 1:1 *v*/*v*, detection: anisaldehyde/sulfuric acid and heating).

### 3.3. Counter-Current Chromatography (CCC) Isolation of Main Compounds

The CCC separation of the main compounds from Sephadex Fractions 8–15 was performed on a *J*-type Quattro MK5 Lab Prep equipment (AECS, Wales, UK). The CCC is equipped with four PTFE coils of 5 m × 2.16 mm (ID), with a total column volume of 500 mL and 120 mm of revolution radius. A manual sample injection valve with a 10 mL loop was used to introduce the sample into the column. The injection volume was 10 mL. The choice of the proper biphasic system for the CCC separation was carried out according to Reference [14]. Several biphasic solvent systems were tested to obtain suitable partition ratios (K*_D_*). The solvent systems were based on different combinations of tert-butylmethylether (TBME), 1-butanol, acetonitrile (ACN), and water acidified with 0.1% of trifluoroacetic acid (TFA) (TBME/BuOH/ACN/H_2_O) in different ratios. For the evaluation, approximately 5 mg of the sample was dissolved in a test tube containing 4 mL of the thoroughly equilibrated solvent systems to be tested (1:1, v/v). The test tubes were shaken, and the compounds were allowed to partition between the two phases. After the equilibration, the phases were separated and analysed by TLC in ALUGRAM^®^ plates (Macherey-Nagel GmbH & Co, Düren, Germany), using as mobile phase EtOAc:formic acid:H_2_O:acetone (10:2:3:1 *v*/*v*/*v*/*v*), and revealed with *p*-anisaldehyde-sulfuric acid. The K*_D_* value of the main compound was calculated using a densitometric analysis of the TLC spots in the superior and inferior phases using the Plot Spot function of ImageJ software (ver. 1.50i, NIH, Bethesda, MD, USA). Briefly, TLC plates were photographed and the areas-under-the-curve (AUCs) of the pixels observed in the TLC spots were determined. The K*_D_* values were calculated, dividing the AUC from the organic phase vs. the AUC from the aqueous phase, in order to carry out separations in the head-to-tail mode. The K*_D_* values are summarized in Table 1. The day of the experiment, 2 L of the selected biphasic solvent system were prepared and separated in a separatory funnel, and then degassed by ultra-sonication for 15 min. The system was filled with the upper stationary phase using a HPLC pump (Serie II, Scientific Systems Inc., State College, PA, USA). The lower aqueous mobile phase was pumped at a flow rate of 4.0 mL/min. Rotation speed was then set at 680 rpm and temperature was maintained at 25 °C during all separations. After reaching hydrodynamic equilibrium, the sample was dissolved in 10 mL of a 1:1 *v*/*v* mixture of the upper and lower phase, sonicated, filtrated through a Clarinert Syringe filter 0.45 μm, 25 mm (Agela Technologies, Torrance, CA, USA), and manually injected into the CCC system. The injection volume was 10 mL. Fractions were collected using a Gilson FC 203B (company, Middleton, WI, USA) set at 0.8 min/tube and pooled after TLC comparison with the system described above. A total of 80 fractions were collected. To end the CCC, rotation was stopped, and the column content was extruded with MeOH at 6 mL/min. The separation by CCC was carried out two times, with a total mass injection of 1 g.

### 3.4. HPLC-DAD-MS/MS^n^ Analysis 

The extracts were compared by HPLC-DAD to obtain the profiles of the plant stem and leaves and to set-up the conditions for HPLC-MS/MS analysis. The hydroalcoholic extracts from stem and roots of *A. marginata* were analyzed using a Shimadzu Prominence chromatographer (Shimadzu Corporation, Kyoto, Japan). The HPLC was equipped with a LC-20AT pump, CTO-20AC column oven, and SPD-M20A UV diode array detector, operated by Labsolution software (ver. 5.6, Shimadzu Corp., Kyoto, Japan)). A MultoHigh 100 RP 18-5 µm (250 × 4.6 mm) column (Cs-Chromatographie Service GmbH, Langerwehe, Germany) was used. The column oven was set at 25 °C. Approximately 10 mg of each extract was dissolved in 1 mL MeOH, filtered through a 0.45 µm PTFE filter, and injected. The gradient solvent system consisted of H_2_O and formic acid (99:1, *v*/*v*, solvent A) and acetonitrile (ACN, solvent B). The initial composition was 10% B. The gradient was started with 90% A and 10% of B. The solvent ratio was then changed as follows: at min 15, 85% A and 15% B; at min 20, 85% A and 15% B; at min 25, 82% A and 18% B; at min 50, 82% A and 18% B; at min 80, 70% A and 30% B. The solvent ratio was then returned to the initial conditions and the column was stabilized for an additional 10 min before the next injection. The flow rate was 0.5 mL/min and the injected volume was 20 μL. The compounds were monitored at 280 nm. 

Mass spectra were recorded using an Agilent 1100 (Agilent Technologies Inc., Santa Clara, CA, USA) liquid chromatography system connected through a split to an Esquire 4000 Ion Trap LC/MS^n^ system (Bruker Daltonics, Bremen, Germany). Ionization was performed at 3000 V, assisted by nitrogen as nebulizing gas at 50 psi and as drying gas at 365 °C with a flow rate of 10 L/min. Negative ions were detected using full scan (*m*/*z* 20–2200) and normal resolution (scan speed 10,300 *m*/*z*/s; peak with 0.6 FWHM/*m*/*z*). The trap parameters were set in the ion charge control (ICC) using manufacturer default parameters, and a maximum accumulation time of 200 ms. The mass spectrometric conditions for the positive ion mode were as follows: electrospray needle, −2500 V; skimmer 1, 19.2 V; skimmer 2, 5.7 V; capillary exit offset 1, 33.0 V; capillary exit offset 2, 52.1 V. The mass spectrometric conditions for the negative ion mode were as follows: electrospray needle, 3500 V; skimmer 1, 20.3 V; skimmer 2, 6.0 V; capillary exit offset 1, 68.2 V; capillary exit offset 2, 88.5 V. The scan mode was performed at a speed of 13,000 *m*/*z*/s, in the range of 50 to 1200 *m*/*z*. Collision-induced dissociation (CID) spectra were obtained with a fragmentation amplitude of 1.00 V (MS/MS) using helium as the collision gas, and were automatically controlled through SmartFrag option. 

### 3.5. Inhibition of Phosphodiesterase (PDE-5)

The inhibition of PDE-5 was carried out following the manufacturer’s instructions of the kit (ab139460, Abcam, Cambdrige, UK), as described in Reference [12]. Crude extracts from stem and roots, as well as the main isolated compounds, were evaluated in concentrations ranging from 0–500 μg/mL. Sildenafil citrate (Europharma, Sao Paulo, Brazil) was used as the positive control for inhibition of PDE-5. Absorbance was read at 600 nm in a microplate reader (ELx800, Biotek, Winooski, VT, USA). The results are expressed as the concentration of sample that inhibited PDE-5 by 50% (IC_50_, μg/mL). 

## 4. Conclusions

Chemical profiling of the Paraguayan crude drug ysypó hû (*A. marginatum*) allowed the tentative identification of 20 constituents, including 9 iridoids, 6 phenolics, and 5 nitrogen-containing compounds. The main compounds were theviridoside and 4-hydroxy-1-methyl-L-proline. In addition, the presence of four chlortheviridosides is reported here for the first time. The present study showed that the crude drug and main compound were inactive towards PDE-5 in vitro. Nevertheless, the potential of ysypó hû as a male sexual enhancer cannot be discarded, since other mechanisms may be involved. Further in vitro and in vivo investigations are still needed to validate or refute the traditional use of ysypó hû as a male sexual enhancer.

## Figures and Tables

**Figure 1 molecules-25-00180-f001:**
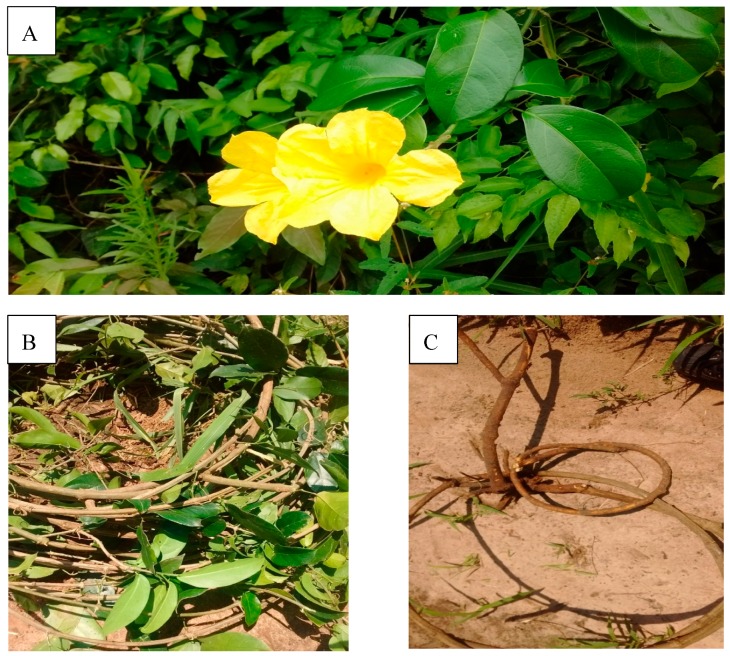
The Paraguayan crude drug “ysypó hû” (*Adenocalymma marginatum*). (**A**) flowering plant; (**B**) aerial parts; (**C**) root and stem.

**Figure 2 molecules-25-00180-f002:**
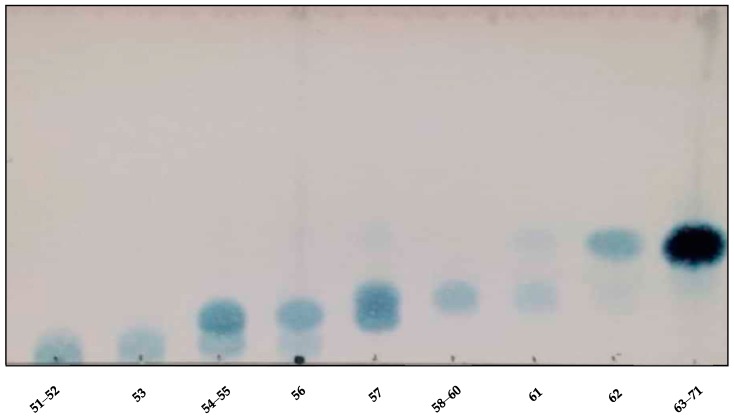
TLC of the counter-current chromatography (CCC) fractions of the Sephadex 8–15 of *Adenocalymma marginatum* (silica gel, EtOAc: formic acid: H2O: acetone 10:2:3:1).

**Figure 3 molecules-25-00180-f003:**
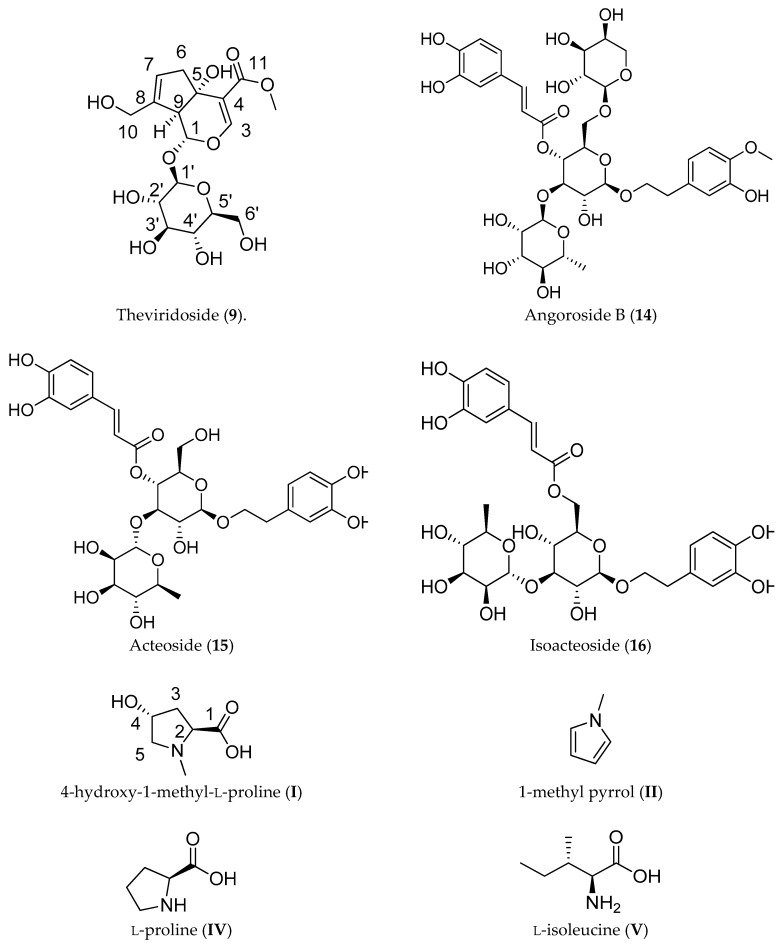
Structure of some of the main constituents of the Paraguayan crude drug ysypó hû (*Adenocalymma marginatum*). Arabic and roman numbers in parenthesis refer to Table 4.

**Figure 4 molecules-25-00180-f004:**
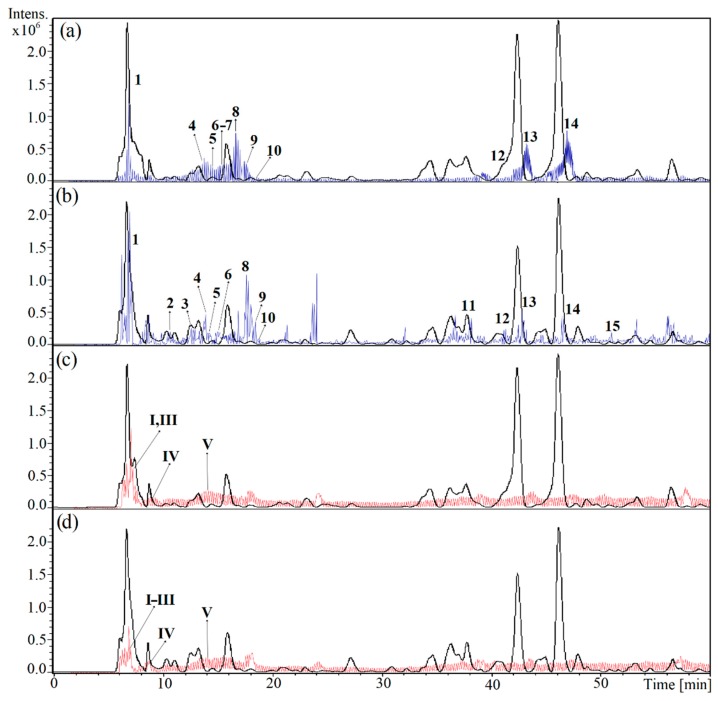
Ion chromatogram in the negative ionization mode (blue trace) and positive ionization mode (red trace), and HPLC profile at 280 nm of the root (**a**,**c**) and stem (**b**,**d**) extracts from the Paraguayan crude drug ysypó hû. Numbers refer to Table 4.

**Table 1 molecules-25-00180-t001:** Summary of the partition ratios (*K*_D_) values of biphasic solvent system (TBME/1-BuOH/ACN/H_2_O) for the main compounds in the Sephadex 8–15 fractions from the hydroalcoholic stem extract of *Adenocalymma marginatum*.

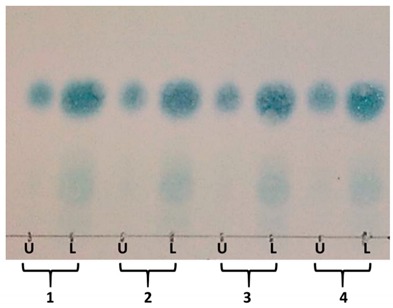		
System Number	Proportions of TBME:*n*-BuOH:ACN:H_2_O +0.1% TFA	*K*_D_ Main Compound
1	1:3:1:5 *	0.21
2	1:4:1:5	0.33
3	1:5:1:5	0.41
4	2:2:2.5:5	0.40

* Selected solvent system; U: upper organic phase; L: lower aqueous phase.

**Table 2 molecules-25-00180-t002:** ^1^H, ^13^ C nuclear magnetic resonance (NMR) and heteronuclear multiple bond correlation (HBMC) data of theviridoside (compound **8**) and its pentaacetate (400 and 100 MHz, respectively; MeOH-d4 or CDCl_3_, δ-values, *J* in. Hz).

	^1^H	^13^C	^1^H	^13^C	HMBC
	MeOH-d4	MeOH-d4	CDCl_3_	CDCl_3_	CDCl_3_
1	5.55 d (5.6)	95.67 d	5.32 d (4.4)	95.26 d	
3	7.52 s	152.71 d	7.31 s	151.36 d	166.63, 113.78, 95.26, 74.91
4	-	113.14 s	-	113.78 s	
5	-	75.12 s	-	74.91 s	5.32
6	2.85 br s	45.54 t	2.75 d (19.6) 2.80 d (19.6)	45.54 t	129.81, 134.68, 74.91
7	5.72 s	125.44 d	5.68 br s	129.81 d	
8	-	140.59 s	-	134.68 s	
9	3.06 d (6.0)	55.37 d	3.08 d (2.4)	54.94 d	
10	4.25 d (14.4) 4.14 d (14.4)	59.63 t	4.58 d (13.6) 4.51 d (13.6)	61.44 t	
11	-	166.96 s	-	166.63 s	
OMe	3.75 s	50.41 q	3.67 s	51.41 q	
1’	4.65 d (8.0)	98.66 d	4.73 d (8)	96.51 d	
2’	3.25 dd (8.8, 8.0)	73.18 d	4.91 t (8.8)	70.73 d	
3’	3.33 m	77.03 d	5.16 t (9.6)	72.03 d	
4’	3.42 m	76.18 d	5.00 t (9.6)	68.15 d	
5’	3.32 m	70.15 d	3.67 m	72.10 d	
6’	3.83-3.85 m, 3.60-3.65 m	62.87 t	4.17 dd (12.4, 4) 4.07 dd (12.4, 1)	61.44 t	
Ac					
COO	-	-	-	170.51 s (2C), 170.05 s, 169.76 s, 169.27 s	
CH_3_	-	-	2.00 s, 1.98 s, 1.93 s, 1.91 s, 1.89 s	20.74 q, 20.62 q, 20.51 q (2C), 20.39 q	

**Table 3 molecules-25-00180-t003:** ^1^H- and ^13^C-NMR data of 4-Hydroxy-1-methyl-l-proline (compound **I**)(400 and 100 MHz, respectively; MeOH-d_4_, δ-values, *J* in Hz).

	H	C	HMBC	Configuration
1	-	170.38 s	4.36, 3.08, 2.27	
2	4.36 dd (11.2, 7.6)	69.03 d	3.88, 3.16, 3.08, 2.27	S
3	2.50 ddt (14, 7.2, 1.6), 2.27 ddd (14, 11.2, 4.8)	38.48 t		
4	4.55 m	68.84 d		R
5	3.88 dd (12.4, 4.4), 3.16 dt (12.4, 2)	63.29 t	3.08	
N-CH_3_	3.08 s	42.79 q	3.88, 4.36; 63.29, 68.84	

**Table 4 molecules-25-00180-t004:** Tentative identification of compounds from root and stem hydroalcoholic extracts of *Adenocalymma marginatum* by HPLC-ESI-MS/MS in the negative ion mode and distribution in roots and stems.

Peak	Rt (min)	[M + H]^+^/[M − H]^−^ *m*/*z*	MS/MS (%)	Tentative Identification	Roots	Stems
		[M + H]^+^				
I	6.0	145.9	99.50 (100)	4-Hydroxy-1-methyl-l-proline *	X	X
II	6.0	99.6	81.5(100)	1-methyl-2,3-dihydro-3-hydroxypyrrol	-	X
III	6.1	81.5		1-methylpyrrol	X	X
IV	6.4	115.7	70.0(100), 28.1 (23), 43.1(20)	l-proline	X	X
V	15.3	131.9	86.1 (100), 69.1(45), 56.9(4)	l-isoleucine *	X	X
		[M − H]^−^				
1	5.5	340.9	178.4(100)	Caffeoyl hexoside	X	X
2	9.0	727.2	564.8(100), 341(14), 240.5(3)	Theviridoside dihexoside 1	-	X
3	11.2	763.3	727.0 (100), 484.9 (29), 383.1 (43), 340.7 (29), 240.6 (11)	Chlortheviridoside dihexoside derivative	-	X
4	12.3	601.4	564.9 (100), 385.1 (73), 301 (27), 240.6 (4), 222.6 (15)	Chlortheviridoside hexoside derivative	X	X
5	12.6	726.9	502.5(42), 385.1 (73), 240.5(3)	Theviridoside dihexoside 2	X	X
6	13.5	601.2	564.9 (100), 385.1 (13), 240.7 (5)	Chlortheviridoside hexoside derivative	X	X
7	13.6	659.1	402.3(93), 254.9(78), 240.6(100)	Methyl-theviridoside aglycon theviridoside	X	-
8	15.7	403.4	240.67(100), 222.60(21)	Theviridoside*	X	X
9	15.9-16.2	600.9	565.0 (100), 240.8 (13)	Chlortheviridoside hexoside derivative	X	X
10	16.1-16.3	565.5	240.65(100)	Theviridoside hexoside	X	X
11	39.4	625.6	461.2(100), 315.3(3)	Dihydro acteoside	-	X
12	41.3	769.6	607.3(100), 461.2(2)	Angoroside B	X	X
13	41.7–42.1	623.6	477.1(4), 461.2(100), 315.2(2)	Acteoside	X	X
14	43.8–45.5	623.5	461.1(100), 315.2(2), 178.6(3)	Isoacteoside	X	X
15	51.3	515.7	352.9(100), 172.6(7)	Dicaffeoylquinic acid	-	X

* Identity confirmed by NMR spectroscopy analysis—not detected.

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
