# Peer review of "Iridoids and Amino Acid Derivatives from the Paraguayan Crude Drug Adenocalymma marginatum (ysypó hû)"

_molecules, 2020, doi:10.3390/molecules25010180_

Round 1
Reviewer 1 Report
too many references for experimental article. should be reduced.
Dicaffeoylquinic acid should be defined - which isomer, 1,5 or 3,5? or other?
line 359 - the concentration od formic acid should be specified
injection volume should be specified.
why 280 nm was chosen for detection (Amax of some compounds or compromise?)
Reviewer 2 Report
The paper of Schmeda-Hirschmann et al. “Iridoids and Amino Acid Derivatives from the Paraguayan Crude Drug Adenocalymma marginatum (ysypó hû)” aimed to HPLC-MS profiling of Adenocalymma marginatum plant and characterizing selected compounds. I would like to make some remarks that have arisen after reading the manuscript.
In Introducing part (line 56) you use the following description “…the stem and leaves are powdered …”. Why do you use the stems and roots as objects of your research? Did you realize some preliminary study demonstrated the effectiveness of stems and roots instead of leaves? I can’t understand the idea of only two compounds, theviridoside and 4-hydroxy-1-methyl-L-proline, isolation as pure compounds. Theviridoside was previously found in Adenocalymma marginatum [Von Poser, et al. Biochem. Syst. Ecol. 2000, doi: 10.1016/S0305-1978(99)00076-9]. MSn identification allows identifying both compounds using known literature data. Moreover, the bioactivity data (Section 2.3) demonstrative their ineffectiveness. If you want to isolate the main compounds, you have to isolate acteoside and isoacteoside have the greatest peak areas (accordingly to Figure 4 data). Figure 4. The final version of Figure 4 is absolutely inappropriate to publication due to a lot of confusing moments as (1) numeration incomprehensibility; (2) the absence of TIC and DAD mode designation (like TIC as blue/red line, DAD as black line); (3) why TIC peaks runes before DAD peaks (it looks like the DAD detector settled after MS detector which is impossible). Table 4. (1) the name of the third column singed as “m/z” should be corrected (maybe as [M+H]+ / [M-H]-, m/z); (2) I was confused by the description of compound 11 as “[M-H+HCOOH]- theviridoside”. Did you mean “theviridoside isomer” or …? (because compound 9 was identified as theviridoside). Section 2.2.1. Line 174-175. “All [M-H]- ions showed two peaks separated by two amu in a 3:1 ratio, characteristic of the presence of chlorine.” I have no idea, what did you mean and what that's related to. Compounds 12, 8, 3, 4, 6, 10 or 3 and 4? What about including the spectra examples in supplementary material? (not all just selected). Section 2.3. It turns out that the neither Adenocalymma marginatum extract nor two compounds have no effect as phosphodiesterase-5 inhibitors. I think it's only right if you will pay more attention to the chemical aspects of Adenocalymma marginatum. I suppose it could be quantitative data about selected compounds content in plant organs or something else. Section 3.3. I was confused by the absence of Supplementary materials mentioned in the manuscript as Table S1 (line 337) (or you mean Table 1).Author Response
Please see the attachment

Reviewer 3 Report
It is an interesting paper in which the authors verified the validity of the use of the family Bignoniaceae species Adenocalymma marginatum in Guarani medicine as a male sexual enhancer. For this purpose, they used both modern professional analytical techniques (CCC=Counter-Current Chromatography, HPLC/MS/MS, ESI-MS) and biological studies (PDE-5=phosphodiesterase-5 inhibitory activity). The authors determined chemical composition and biological activity of the extracts from the stems and roots of this species. They isolated theviridoside (iridoid) and 4-hydroxy-1-methylproline as the main constituents. Moreover, they for the first time confirmed the presence of three chlorthevirodise hexoside derivatives, previously unknown to occur in the plant kingdom. They also tentatively identified ten iridoids, six phenolic compounds and 5 amino acids. The obtained results encourage the continuation of studies aimed to explain the mechanism of activity of the extract as a male sexual enhancer.
The paper is worthy of publication in the journal Molecules.
Language assessment
The paper is well written in concise and precise language.
A very few mistakes are listed below:
245 sulphur should be replaced by sulfur 247 Bioactivity studies on 246 Adenocalymma include 261 the improved version of this phrase is: “neither the crude root, stem extracts nor the main compounds presented” 316 sulphuric should be replaced by sulfuricAuthor Response
Please see the attachment

Round 2
Reviewer 2 Report
The corrected manuscript of Schmeda-Hirschmann et al. “Iridoids and Amino Acid Derivatives from the Paraguayan Crude Drug Adenocalymma marginatum (ysypó hû)” was fully corrected accordingly early recommendations. It may be accepted after the minor text editing.